# OPT-BENCH: Evaluating LLM Agent on Large-Scale Search Spaces Optimization Problems

## Abstract

Large Language Models (LLMs) have demonstrated impressive capabilities in solving a wide range of tasks. However, their ability to iteratively optimize complex solutions by learning from previous feedback remains underexplored. To address this gap, we introduce **OPT-BENCH**, a comprehensive benchmark designed to evaluate LLM agents on large-scale search space optimization problems. OPT-BENCH includes 20 real-world machine learning tasks sourced from Kaggle and 10 classical NP problems, providing a diverse and challenging environment for assessing LLMs on iterative reasoning and solution refinement. To facilitate rigorous evaluation, we present **OPT-Agent**, an end-to-end optimization framework that emulates human reasoning by generating, validating, and iteratively improving solutions through the use of historical feedback. Through extensive experiments involving 17 state-of-the-art LLMs from 7 model families, including reasoning models, general models, and open-source models ranging from 3B to 72B parameters, we demonstrate that incorporating historical context significantly enhances optimization performance across both ML and NP tasks. However, this benefit remains limited, as even with the latest models, a gap still persists compared to human expert performance. All datasets, code, and evaluation tools will be open-sourced to foster further research in advancing LLM-driven optimization and iterative reasoning.

## 1 Introduction

The advent of Large Language Models (LLMs) OpenAI (2025); Grattafiori et al. (2024); Guo et al. (2025a) has revolutionized artificial intelligence, demonstrating exceptional performance across a wide range of tasks Brown et al. (2020); Ouyang et al. (2022); Achiam et al. (2023); Chowdhery et al. (2023); Touvron et al. (2023); Google (2024). As LLMs continue to evolve, their potential to solve complex problems through iterative refinement is becoming increasingly apparent. Several benchmarks have been introduced to evaluate LLMs' reasoning abilities, with a focus on tasks such as mathematical problem solving Amini et al. (2019); Fan et al. (2023); Guo et al. (2025b); Cobbe et al. (2021); Lightman et al. (2023), decision-making Valmeekam et al. (2023), and logical reasoning Creswell et al. (2022); Xu et al. (2025).

Despite these advances, an essential aspect of human cognition—learning from both successes and failures over time—remains largely unexplored in current LLM evaluations Hendrycks et al. (2020); Cobbe et al. (2021). Humans routinely refine their reasoning by integrating feedback, as seen when scientists adjust hypotheses, students revise study strategies, or chess players improve tactics based on past outcomes. In contrast, current evaluations of large language models (LLMs) primarily measure their ability to generate correct responses in a single pass Fan et al. (2023); Lin et al. (2024), overlooking the ability to learn from experience and adapt over multiple iterations. Currently, there is a lack of benchmark tasks that explicitly evaluate LLMs' capacity for iterative learning and reasoning based on prior experience.

To address existing limitations, we introduce OPT-BENCH, a comprehensive benchmark designed to evaluate LLM performance on large-scale search space optimization problems. OPT-BENCH comprises 30 diverse tasks, including 20 real-world machine learning (ML) challenges sourced from Kaggle competitions and 10 classical NP-complete (or NP-hard) combinatorial optimization problems. The ML tasks span predictive domains such as regression and classification, including applications like

house price forecasting and sentiment analysis. The NP problems cover core computational challenges from graph theory, scheduling, and resource allocation, including graph coloring, Hamiltonian cycle, and the knapsack problem. These tasks are characterized by combinatorial complexity and the absence of polynomial-time algorithms for large instances, making them ideal for assessing iterative optimization capabilities. Additionally, to evaluate the model's performance relative to human ability, we further curate solutions from human experts. While obtaining the optimal solutions for these problems is challenging, for ML tasks, we gather gold-standard solutions from leaderboards, and for NP problems, we implement heuristic algorithms to approximate optimal solutions.

Furthermore, we propose OPT-Agent, an LLM optimization pipeline that mirrors human problem-solving by supporting a comprehensive, end-to-end workflow for solution generation, validation, and iterative refinement. This framework enables rigorous evaluation of LLMs' ability to optimize solutions over multiple iterations by leveraging historical context and adapting based on feedback. Unlike prior benchmarks that focus on isolated or synthetic tasks, OPT-BENCH emphasizes real-world challenges with extended iteration horizons, pushing the boundaries of LLM optimization capabilities. OPT-Agent generates solutions, incrementally refines them using feedback, and performs debugging informed by previous errors. Distinct from earlier agent frameworks Jiang et al. (2025); Huang et al. (2024); Chan et al. (2024) that assess single-pass solution generation, OPT-Agent emphasizes evaluating the model's capacity to learn from feedback history. Together, OPT-BENCH and OPT-Agent provide a robust platform for advancing research in LLM-driven optimization across both machine learning and combinatorial domains.

Using this benchmark and optimization framework, we evaluate seventeen leading LLMs from seven model families on OPT-BENCH, exploring the impact of iteration count and hyperparameters, such as temperature, on both solution refinement and draft generation for ML tasks. Our results show that incorporating historical context consistently enhances optimization performance across both ML and NP problems, facilitating more effective iterative refinement. Increasing the number of optimization steps further improves outcomes, highlighting the importance of extended iterations for convergence. However, this benefit remains limited, and even with the latest models, a gap persists compared to human expert performance. Temperature plays a critical role in balancing exploration and stability, with lower to moderate values typically yielding optimal performance; however, the optimal settings vary depending on the model and task. Although draft optimization results in higher rates of invalid solutions, it often outperforms traditional refinement methods. Notably, open-source models exhibit higher error rates and lag behind proprietary counterparts on NP tasks, suggesting areas for improvement. Additionally, we conduct a detailed case study to analyze why integrated historical information is less effective for NP problems compared to ML tasks.

In summary, our contributions are as follows:

- We present OPT-BENCH, a benchmark comprising 20 machine learning tasks and 10 NP problems, specifically designed to assess large language models' (LLMs) ability to solve problems with large search spaces. It evaluates whether models can improve solutions over time by learning from past feedback.

- We introduce OPT-Agent, an end-to-end automated evaluation framework that enables LLMs to learn from historical feedback when solving practical, real-world optimization problems, thereby advancing their cognitive capabilities in iterative reasoning and improvement.

- We perform extensive experiments on 17 state-of-the-art LLMs from 7 model families, including reasoning models, general models, and open-source models ranging from 3B to 72B parameters. Our analysis provides insights that can help guide future research on enhancing LLMs' optimization capabilities.

## 2 OPT-BENCH

OPT-BENCH consists of 30 tasks: 20 machine learning challenges from Kaggle competitions and 10 classic NP problems. The ML tasks span a range of predictive problems—including regression, classification, and error prediction—with applications such as sales forecasting, house price estimation, sentiment analysis, and multi-class classification. The NP tasks cover fundamental combinatorial optimization and graph-theoretic problems, including graph coloring, Hamiltonian cycle, set cover, subset sum, knapsack, maximum clique, minimum cut, and traveling salesman (TSP). These prob-

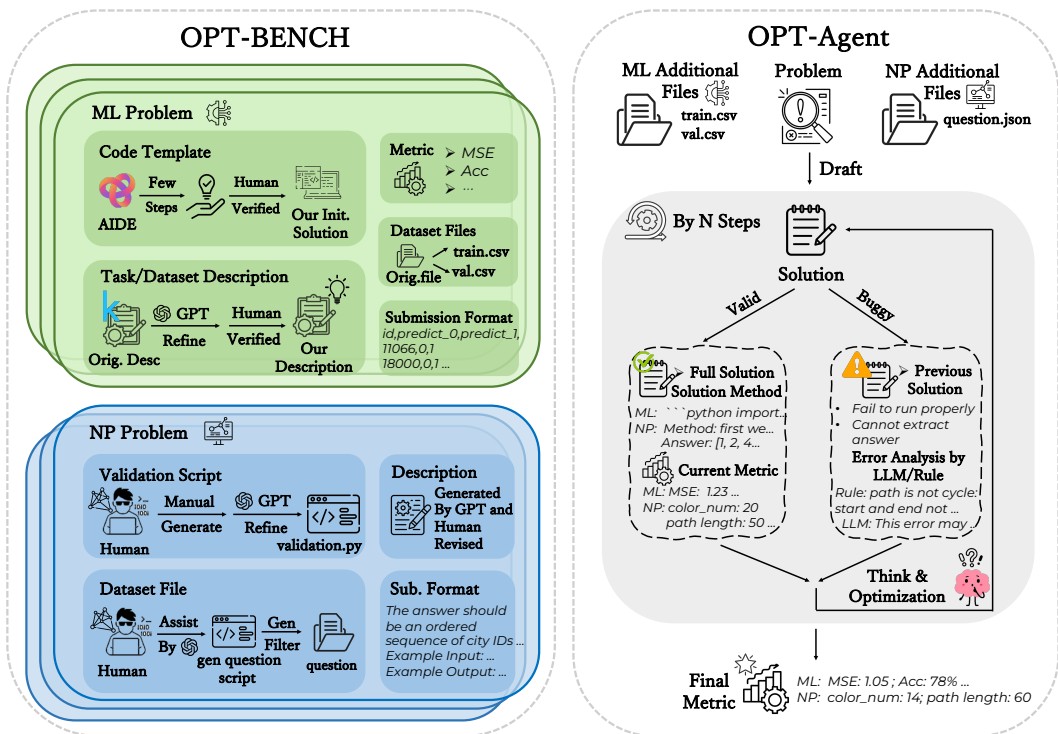

Figure 1: **Overview of the OPT-BENCH dataset and OPT-Agent Framework.** The left panel illustrates the data structure of OPT-Bench, encompassing ML and NP problems. Each module includes problem definitions, dataset files, validation script (NP), evaluation metrics, and submission formats, integrating human-verified initial solutions and LLM-assisted refinement. The right panel details the evaluation workflow, where solutions are iteratively generated and validated in steps. Valid solutions proceed to metric calculation, while buggy solutions trigger error analysis via LLM or rule-based diagnostics, followed by iterative optimization. This framework enables systematic and automated assessment of LLMs' optimization capabilities across diverse problem domains.

lems are notable for their computational intractability at scale due to the lack of polynomial-time algorithms. See Appendix A for the complete task list.

## 2.1 DATASET CURATION AND ANALYSIS

As illustrated in Figure 1, the preparation of ML tasks begins with the collection of well-defined task descriptions from the Kaggle website. These descriptions are subsequently refined using gpt-4o and then meticulously verified by human experts to ensure clarity and conciseness. Each task specifies its evaluation metric to rigorously assess model performance. The corresponding training and test datasets are sourced directly from Kaggle competitions and provided as separate files. In addition, comprehensive dataset descriptions detail the format and features of the training data, facilitating proper model training and evaluation. The submission format prescribed by the Kaggle competition is strictly followed to guarantee consistency throughout the evaluation process. To initialize the solution space, an initial solution for each ML task is generated by the LLM-based machine learning agent AIDE Jiang et al. (2025), which employs a 10-step solution generation process to identify the best candidate. This initial solution is then further refined by four PhD-level experts to ensure correctness and to enhance its potential for subsequent optimization. Furthermore, we gather the gold medal solution from the Kaggle leaderboard to serve as the human expert baseline.

Regarding the NP tasks, the benchmark focuses on 10 classical NP problems, each accompanied by a detailed task description and an explicit goal definition. The expected submission format is clearly outlined to enforce adherence to the required solution structure. To further guide the LLMs, example inputs and outputs are provided, illustrating the correct format and helping to reduce ambiguity

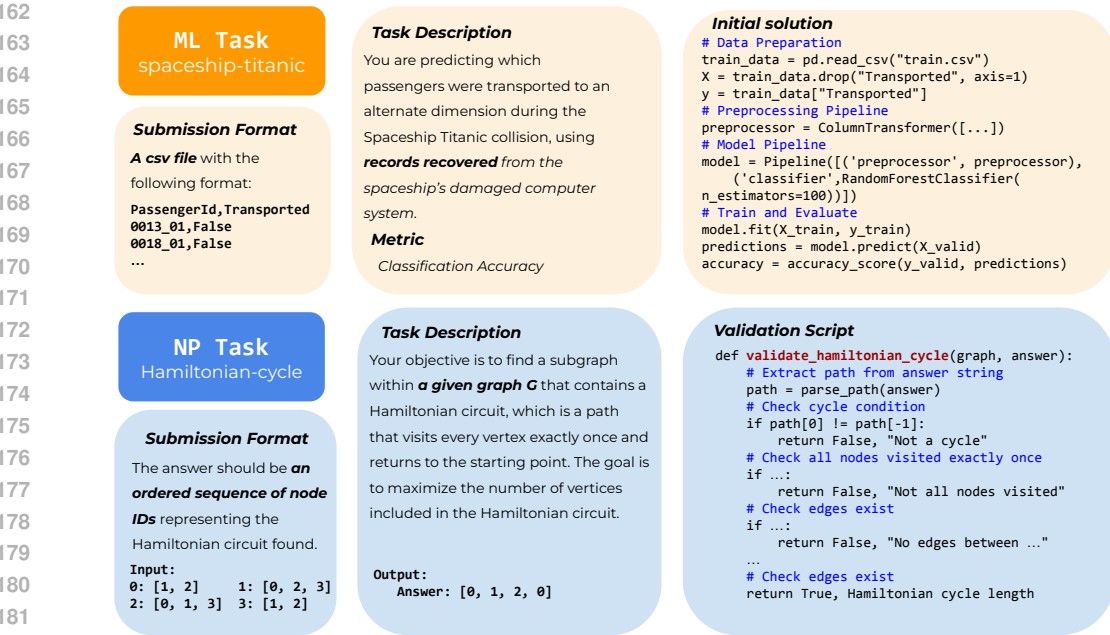

Figure 2: **Specific cases from OPT-BENCH**. Take the spaceship titanic classification task and the Hamiltonian cycle optimization problem as representative examples.

during solution generation. Each NP problem is encapsulated in a JSON file containing five distinct instances, enabling a robust and comprehensive evaluation of model performance. To ensure solution validity, a rule-based Python script `validation.py` is employed. This script rigorously verifies whether the submitted solution complies with the constraints of the respective NP problem. For instance, in the Hamiltonian cycle problem, the script confirms that no vertex is revisited and that the output constitutes a valid cycle. Solutions failing these checks receive an error message, while valid solutions have their performance metrics computed and returned. Furthermore, we apply a heuristic algorithm to find the approximate optimal solution, which serves as the human expert baseline.

Each sample in the OPT-BENCH includes the following elements (see Figure 2):

- Task Description: A description of the ML or NP problem, including the background and the optimization objectives.

- Dataset specifications: ML tasks provide training and test datasets in CSV format, accompanied by detailed descriptions of data format; NP tasks supply problem instances formatted as JSON files, each containing inputs of varying difficulty.

- Submission formats: it defines the requirements for solution submission, making the output easy to extract and further evaluation.

- Initial solution: For ML tasks, an initial solution script is provided to serve as a foundational baseline for further optimization and refinement.

- Evaluation principles and metrics: The ML tasks are evaluated based on performance metrics computed on a designated test set. For NP problems, a rule-based script `validation.py` is provided to automatically verify solution validity and compute corresponding metrics for feasible outputs. The metric for each task is provided in Appendix A.

- Human Expert Annotation: For each task in OPT-BENCH, we curate human expert solutions. Specifically, for ML tasks, we gather the gold medal solutions from the Kaggle leaderboard, and for NP tasks, we apply a heuristic algorithm to find the approximate optimal solution.

Table 1 compares OPT-BENCH with three representative benchmarks, emphasizing differences in task composition, evaluation scope, and metrics. OPT-BENCH includes 20 ML and 10 NP tasks and utilizes multiple metrics offering a more comprehensive and detailed evaluation of LLM optimization.

| Benchmark | Task Num | | Iterative Optimization | Metrics |
|---|---|---|---|---|
| | ML | NP | | |
| NPHardBench Fan et al. (2023) | ✗ | 9 | ✗ | *Weighted Accuracy, Failure Rate* |
| MLE-Bench Chan et al. (2024) | 75 | ✗ | ✓ | *Medal Rate* |
| MLAgentBench Huang et al. (2024) | 13 | ✗ | ✓ | *Success Rate, Avg. Improvement* |
| **OPT-BENCH** | 20 | 10 | ✓ | *Win Count, Buggy Rate, IR, AR* |

Table 1: **Comparison of OPT-BENCH with other representative agent benchmarks.**

This extensive task diversity and multifaceted, iterative assessment establish OPT-BENCH as a robust benchmark for evaluating LLM's optimization capabilities.

## 2.2 AGENT WORKFLOW

In this section, we introduce OPT-Agent, a framework inspired by the AIDE Jiang et al. (2025) and rooted in human cognitive chain-of-thought problem-solving strategies. The approach begins with a simple initial solution and iteratively refines it based on feedback from previous results. To replicate this human-like reasoning, OPT-Agent is designed as an LLM-driven system that models key elements of human problem-solving. As illustrated in Figure 1, OPT-Agent performs three core actions:

- **Drafting**: This action is performed at the beginning. The agent is prompted to generate a new initial solution to the problem. For machine learning tasks, it produces a Python script, whereas for NP problems, it formulates a specific answer to the question at hand.

- **Improving**: This action is triggered when the previous solution is valid. The LLM is prompted to optimize the solution further. For ML tasks, it adjusts model architecture, feature engineering, or hyperparameters to improve performance on the validation set. For NP tasks, it seeks a better solution based on the task description. Additionally, historical information from previous valid solutions, including the code for ML tasks or the answer path for NP tasks, along with corresponding metrics and analyses, is provided to guide the optimization process.

- **Debugging**: This action is invoked when the previous solution contains errors. The LLM is prompted to debug and fix the issues. For ML tasks, it inspects error logs, analyzes mistakes, and corrects issues such as tensor dimension mismatches, feature engineering errors, or other coding mistakes. For NP tasks, the LLM revises the error messages generated by the validation script.

## 3 EXPERIMENTS AND RESULTS

### 3.1 EXPERIMENTAL ENVIRONMENTS

All OPT-BENCH-ML experiments were conducted on an Ubuntu 20.04 system with 4 CPU cores and 32 GB of RAM, while OPT-BENCH-NP experiments were run using 2 CPU cores and 16 GB of RAM. No GPU resources were required. Proprietary LLMs were accessed via API, whereas open-source LLMs were queried through LMDeploy.

### 3.2 EXPERIMENTAL SETTINGS

We evaluate LLM performance on ML and NP tasks from OPT-BENCH using four metrics:

- **Win Count**: Expressed as $x_1/x_2$, where $x_1$ represents the number of tasks where using historical information yields better performance, and $x_2$ denotes the number of tasks where omitting historical information performs better. Both conditions are evaluated under the same experimental settings, with the sole difference being the inclusion of historical information. This metric qualitatively assesses the optimization capability of the LLM in our benchmark.

- **Buggy Rate**: The proportion of tasks for which the LLM fails to generate a valid solution. It serves as an indicator of model robustness and reliability during the optimization process.

- **Average Ratio (AR)**: The average ratio of each metric partition across all tasks is compared to the human expert baseline, providing a measure of the model's solution quality relative to human expert solution.

- **Improvement Rate (IR)**: A quantitative measure designed to evaluate the optimization capability of LLMs under different experimental settings, primarily used in OPT-BENCH-ML. It's defined as:

$$\text{IR}(\alpha, \beta) = \frac{1}{n} \sum_{i=1}^{n} \frac{\alpha_i}{\beta_i},$$

where $\alpha_i$ and $\beta_i$ denote the performance metrics (e.g., MSE or accuracy) for the $i$-th ML task, and $n = 20$ is the total number of tasks. Typically, $\alpha_i$ corresponds to the metric variable value in the improved setting, while $\beta_i$ represents either the variable value under baseline setting or the metric from initial solution. This metric quantitatively reflects the relative performance improvement achieved under varying optimization configurations.

## 3.3 MAIN EXPERIMENT

To assess the effect of historical information on LLM optimization, we compare OPT-Agent with and without historical context (baseline) across optimization steps ranging from 5 to 20.

**OPT-BENCH-ML.** The results shown in Table 2 reveal several key findings. First, incorporating historical information consistently improves optimization performance across most models, as indicated by Win Count and improvement rates *IR(w,w.o)* greater than one, highlighting the value of contextual information for iterative solution refinement. Second, the significant upward trend in *IR(w,init)* with increasing steps demonstrates that longer optimization horizons generally lead to better performance, allowing for more thorough exploration and convergence. Notably, gpt-o3-mini achieves the highest overall improvement relative to the initial solution across all step counts. Third, for models such as gemini-2.0-flash and DeepSeek-V3.1-Thinking, *IR(w,w.o)* decreases as the number of steps increases from 10 to 20, suggesting that some LLMs fail to effectively utilize long context information, which is a key aspect for future improvement. Additionally, upon analyzing *AR*, we conclude that for closed-source models, good solutions are typically found in the early steps, with minimal improvement from 5 steps to 20 steps. However, for open-source models, although they start with worse solutions, optimization efficiency increases up to 20 steps, as observed in Qwen3-32B. Finally, even the best proprietary models still exhibit a performance gap when compared to human experts.

**OPT-BENCH-NP.** Table 3 shows that models generally perform better on NP problems when historical information is incorporated, mirroring trends observed in ML tasks. The *Buggy Rate* metric, specific to NP tasks, highlights differences in solution validity across models and iterations. Most models exhibit a reduction in *Buggy Rate* with the inclusion of historical information, with gpt-o3-mini achieving zero errors, demonstrating the benefit of using error feedback for iterative correction. The *Rank* metric indicates that lower buggy rates do not always correspond to higher optimization ranks, suggesting that some models prefer valid but suboptimal solutions. While increased optimization steps generally improve performance and reduce errors, these gains are often nonlinear; for instance, gemini-2.0-flash and Qwen2.5-72B-Instruct show significant improvement with more iterations, while others plateau beyond 10 steps, indicating that optimal step counts vary by model. Notably, the open-source Qwen2.5-72B-Instruct underperforms compared to proprietary models, pointing to ongoing challenges and opportunities for improvement in open-source LLMs for NP problems. Additionally, reasoning models significantly outperform general models of the same size (e.g., Qwen3-8B vs. Qwen2.5-7B-Instruct). For reasoning models, the performance often starts strong at step 5, with fewer improvements in subsequent steps. In contrast, general models show significant improvements as the number of optimization steps increases.

## 3.4 ABLATION STUDY

To further investigate the influence of temperature on LLM performance, we conduct temperature experiments on general LLMs, as reasoning models often require specific temperature settings. Additionally, we perform OPT-Agent-draft experiments on ML tasks, with detailed results provided in the Appendix. **OPT-BENCH-ML** Results shown in Table 4 indicate that optimal temperature varies across models. For example, gpt-4o-2024-08-06 attains its highest win count and improvement rates, and grok-3 achieves the best *IR(w,init)* at temperature 0, suggesting that more deterministic decoding enhances stability and consistency in optimization. Conversely, increasing temperature to 0.8 generally leads to a decline in both win counts and improvement metrics for proprietary models, implying that excessive randomness may impair optimization robustness and the effective

| Model | 5 steps | | | 10 steps | | | 20 steps | | |
|---|---|---|---|---|---|---|---|---|---|
| | Win Count | IR(w,w.o) | AR | Win Count | IR(w,w.o) | AR | Win Count | IR(w,w.o) | AR |
| *Proprietary LLMs* | | | | | | | | | |
| gpt-4o-2024-08-06 | 13/7 | 1.28 | 0.53 | 13/7 | 1.80 | 0.58 | **18/2** | 1.89 | 0.61 |
| gpt-4.1-2025-04-14 | 12/8 | 1.11 | 0.40 | 14/6 | 1.67 | 0.50 | 14/6 | **2.15** | 0.58 |
| gpt-o3-mini | 12/8 | 1.25 | **0.55** | 14/6 | 1.77 | **0.63** | 13/7 | 1.35 | **0.65** |
| gemini-2.0-flash | 13/7 | 1.08 | 0.45 | 14/6 | **2.08** | 0.48 | 14/6 | 1.95 | 0.53 |
| claude-3-5-sonnet-20241022 | 11/9 | 1.15 | 0.30 | 12/8 | 1.45 | 0.40 | 12/8 | 1.83 | 0.48 |
| claude-3-7-sonnet-20250219 | 14/6 | 1.00 | 0.43 | 14/6 | 1.11 | 0.48 | 14/6 | 1.35 | 0.53 |
| grok-3 | 12/8 | 1.16 | 0.50 | 14/6 | 1.36 | 0.55 | 15/5 | 1.29 | 0.63 |
| Deepseek-V3.1-Thinking | 12/8 | 1.13 | 0.50 | 12/8 | 1.18 | 0.58 | 11/9 | 1.03 | 0.60 |
| *Open-Source LLMs* | | | | | | | | | |
| Internlm3-8b-instruct | 8/12 | 0.94 | 0.05 | 9/11 | 0.98 | 0.23 | 10/10 | 1.01 | 0.30 |
| Qwen2.5-3B-Instruct | 8/12 | 0.98 | 0.05 | 7/13 | 0.85 | 0.13 | 6/14 | 0.63 | 0.15 |
| Qwen2.5-7B-Instruct | 9/11 | 0.99 | 0.03 | 7/13 | 0.82 | 0.15 | 6/14 | 0.63 | 0.20 |
| Qwen2.5-14B-Instruct | 10/10 | 1.39 | 0.18 | 11/9 | 1.11 | 0.25 | 12/8 | 1.13 | 0.30 |
| Qwen2.5-32B-Instruct | 9/11 | 1.38 | 0.25 | 10/10 | 1.21 | 0.33 | 11/9 | 1.19 | 0.45 |
| Qwen2.5-72B-Instruct | **15/5** | **1.58** | 0.40 | **15/5** | 1.47 | 0.43 | 15/5 | 1.61 | 0.45 |
| Qwen3-8B | 9/11 | 0.99 | 0.28 | 10/10 | 1.24 | 0.33 | 12/8 | 1.16 | 0.38 |
| Qwen3-30B-A3B | 10/10 | 1.06 | 0.45 | 11/9 | 1.08 | 0.53 | 12/8 | 1.14 | 0.53 |
| Qwen3-32B | 10/10 | 1.07 | 0.18 | 11/9 | 1.19 | 0.48 | 12/8 | 1.28 | 0.58 |

Table 2: Evaluation Results of LLMs on OPT-BENCH-ML, comparing both closed-source and open-source models, including general and reasoning models.

| Model | 5 steps | | | | | 10 steps | | | | | 20 steps | | | | |
|---|---|---|---|---|---|---|---|---|---|---|---|---|---|---|---|
| | Win Count | Buggy Rate | | AR | | Win Count | Buggy Rate | | AR | | Win Count | Buggy Rate | | AR | |
| | | w | w.o | w | w.o | | w | w.o | w | w.o | | w | w.o | w | w.o |
| *Proprietary LLMs* | | | | | | | | | | | | | | | |
| gpt-4o-2024-08-06 | 5/5 | 0.30 | 0.34 | 0.41 | 0.40 | 4/6 | 0.26 | 0.30 | 0.44 | 0.44 | 4/6 | 0.22 | 0.18 | 0.47 | 0.54 |
| gpt-4.1-2025-04-14 | 4/6 | 0.10 | 0.10 | 0.75 | 0.77 | 4/6 | 0.08 | 0.08 | 0.77 | 0.78 | 4/6 | 0.04 | 0.02 | 0.83 | 0.85 |
| gpt-o3-mini | 5/5 | **0.00** | **0.00** | 0.92 | 0.88 | 6/4 | **0.00** | **0.00** | 0.92 | 0.89 | 5/5 | **0.00** | 0.00 | 0.93 | 0.89 |
| gemini-2.0-flash | 5/5 | 0.16 | 0.20 | 0.40 | 0.42 | 6/4 | 0.06 | 0.12 | 0.44 | 0.43 | **7/3** | 0.06 | 0.10 | 0.45 | 0.40 |
| claude-3-5-sonnet-20241022 | 7/3 | 0.20 | 0.32 | 0.49 | 0.44 | 7/3 | 0.18 | 0.30 | 0.51 | 0.46 | **7/3** | 0.18 | 0.28 | 0.52 | 0.47 |
| claude-3-7-sonnet-20250219 | **8/2** | 0.10 | 0.24 | 0.62 | 0.52 | **8/2** | 0.08 | 0.22 | 0.64 | 0.55 | 7/3 | 0.06 | 0.12 | 0.64 | 0.58 |
| grok-3 | 5/5 | 0.06 | 0.04 | 0.72 | 0.75 | 5/5 | 0.00 | 0.00 | 0.77 | 0.76 | 5/5 | 0.00 | 0.00 | 0.82 | 0.78 |
| Deepseek-V3.1-Thinking | 6/4 | **0.00** | **0.00** | **0.94** | **0.96** | 5/5 | 0.00 | 0.00 | **0.95** | **0.96** | 5/5 | 0.00 | 0.00 | **0.96** | **0.96** |
| *Open-Source LLMs* | | | | | | | | | | | | | | | |
| Internlm3-8b-Instruct | 4/6 | 0.52 | 0.54 | 0.22 | 0.24 | 6/4 | 0.30 | 0.42 | 0.35 | 0.31 | 6/4 | 0.18 | 0.20 | 0.44 | 0.40 |
| Qwen2.5-3B-Instruct | 6/4 | 0.64 | 0.80 | 0.12 | 0.03 | 7/3 | 0.46 | 0.68 | 0.18 | 0.09 | 8/2 | 0.38 | 0.58 | 0.21 | 0.14 |
| Qwen2.5-7B-Instruct | 6/4 | 0.54 | 0.70 | 0.22 | 0.14 | 7/3 | 0.38 | 0.54 | 0.31 | 0.23 | 6/4 | 0.24 | 0.28 | 0.38 | 0.37 |
| Qwen2.5-14B-Instruct | 5/5 | 0.40 | 0.60 | 0.31 | 0.22 | 5/5 | 0.24 | 0.28 | 0.41 | 0.39 | 6/4 | 0.18 | 0.18 | 0.47 | 0.43 |
| Qwen2.5-32B-Instruct | 5/5 | 0.42 | 0.44 | 0.33 | 0.35 | 5/5 | 0.30 | 0.28 | 0.41 | 0.43 | 6/4 | 0.20 | 0.20 | 0.52 | 0.46 |
| Qwen2.5-72B-Instruct | 5/5 | 0.40 | 0.50 | 0.33 | 0.30 | 5/5 | 0.32 | 0.38 | 0.39 | 0.38 | 4/6 | 0.22 | 0.32 | 0.47 | 0.44 |
| Qwen3-8B | 5/5 | 0.28 | 0.22 | 0.61 | 0.66 | 5/5 | 0.20 | 0.20 | 0.67 | 0.68 | 5/5 | 0.12 | 0.20 | 0.73 | 0.70 |
| Qwen3-30B-A3B-Instruct-2507 | 5/5 | 0.04 | 0.12 | 0.80 | 0.79 | 5/5 | **0.00** | 0.06 | 0.85 | 0.85 | 6/4 | **0.00** | **0.00** | 0.86 | 0.92 |
| Qwen3-32B | 5/5 | 0.14 | 0.12 | 0.72 | 0.77 | 4/6 | 0.10 | 0.10 | 0.77 | 0.79 | 5/5 | 0.06 | 0.10 | 0.80 | 0.80 |

Table 3: Evaluation Results of LLMs on OPT-BENCH-ML, comparing both closed-source and open-source models, including general and reasoning models.

use of historical context. Notably, the open-source model `Qwen2.5-72B-Instruct` shows stable performance across temperature settings, with a slight improvement at higher temperatures, suggesting a different sensitivity to sampling variability, likely due to model architecture or training. This finding highlights the importance of tuning temperature to balance exploration and exploitation in LLM-based optimization.

**OPT-BENCH-NP** For NP problems, decoding temperature affects solution validity differently than in ML tasks. As shown in Table 5, lower temperatures lead to higher buggy rates, reducing valid solutions; for instance, `gpt-4o-2024-08-06` shows an increase from 0.18 at 0.2 to 0.28 at temperature 0, indicating that moderate temperatures better balance exploration and reliability. Win count remains stable across temperatures, demonstrating the robustness of historical information.

| Model | Temperature=0 | | | Temperature=0.2 | | | Temperature=0.8 | | |
|---|---|---|---|---|---|---|---|---|---|
| | Win Count | IR(w,w.o) | AR | Win Count | IR(w,w.o) | AR | Win Count | IR(w,w.o) | AR |
| gpt-4o-2024-08-06 | **13/7** | **1.58** | 0.58 | 9/11 | 1.19 | 0.50 | 10/10 | 1.10 | 0.40 |
| grok-3 | 9/11 | 1.01 | 0.65 | **11/9** | **1.29** | 0.45 | 8/12 | 1.04 | 0.53 |
| Qwen2.5-72B-Instruct | 10/10 | 1.05 | 0.43 | 11/9 | 1.04 | 0.40 | **11/9** | **1.11** | 0.44 |

Table 4: Evaluation Results of LLMs on OPT-BENCH-ML Across Different Temperature Settings.

| Model | Temperature=0 | | | | | Temperature=0.2 | | | | | Temperature=0.8 | | | | |
|---|---|---|---|---|---|---|---|---|---|---|---|---|---|---|---|
| | Win Count | Buggy Rate | | AR | | Win Count | Buggy Rate | | AR | | Win Count | Buggy Rate | | AR | |
| | | w | w.o | w | w.o | | w | w.o | w | w.o | | w | w.o | w | w.o |
| gpt-4o-2024-08-06 | **5/5** | 0.28 | 0.24 | 0.48 | 0.44 | **5/5** | 0.18 | 0.28 | **0.51** | 0.47 | 4/6 | **0.18** | 0.22 | 0.47 | **0.49** |
| grok-3 | 4/6 | **0.04** | **0.08** | **0.74** | **0.72** | 5/5 | **0.02** | **0.04** | **0.76** | **0.73** | 4/6 | 0.00 | **0.00** | **0.80** | 0.81 |
| Qwen2.5-72B-Instruct | **5/5** | 0.30 | 0.34 | 0.47 | 0.46 | 4/6 | 0.24 | 0.26 | 0.44 | 0.46 | 5/5 | 0.25 | 0.26 | 0.46 | 0.44 |

Table 5: Evaluation Results of LLMs on OPT-BENCH-NP across Different Temperature Settings.

These results highlight temperature as a key hyperparameter, with moderate settings providing an optimal balance between solution validity and search diversity.

## 3.5 FURTHER DISCUSSION

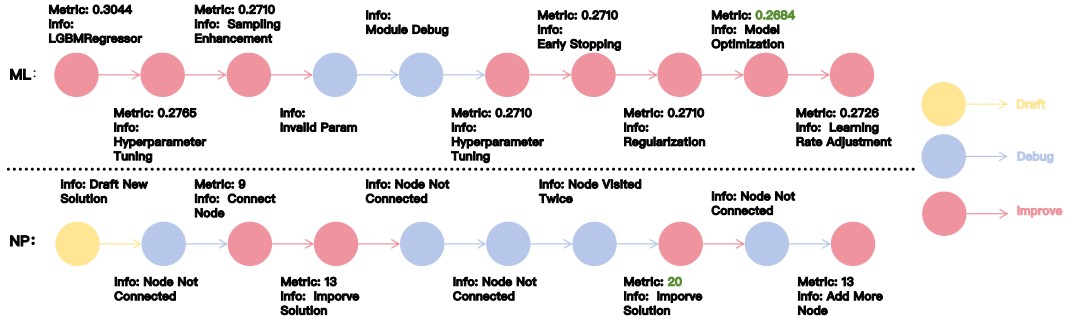

Figure 3: **OPT-Agent Optimization Trace on ML and NP Tasks**. Nodes are color-coded by task status: yellow for Draft, blue for Debug, and red for Improve. Each node displays performance metrics and descriptive details, reflecting iterative improvements during optimization. OPT-Agent leverages historical information to enhance solution quality across ML and NP tasks, exemplified by the bike sharing demand prediction and Hamiltonian cycle problems, respectively.

### 3.5.1 HOW HISTORICAL INFORMATION EFFECTIVE IN OPT-AGENT?

As shown in Figure 3, OPT-Agent leverages historical feedback to optimize the reasoning trace. For ML tasks, valid previous nodes guide improvements in hyperparameters, regularization, learning rates, model structure, and feature engineering, while buggy nodes trigger debugging based on errors like missing packages, feature dimension mismatches, or incorrect parameters. For NP tasks, valid nodes inform solution refinement, and buggy nodes prompt debugging using rule-based information such as repeated visits or disconnected nodes.

### 3.5.2 WHY HISTORICAL INFORMATION IS LESS EFFECTIVE FOR NP TASKS?

As analyzed in Section 3.3, experimental results show that historical information is less effective for OPT-Agent on NP tasks compared to ML tasks. This is mainly because the LLM struggles to interpret and apply feedback incrementally in NP problems. For instance, in the Hamiltonian cycle problem, when OPT-Agent's solution fails due to node A and B not being connected, a human would refine the solution by removing disconnected nodes A and B, whereas the LLM often generates a completely new solution instead of building upon prior feedback. In contrast, ML tasks benefit from more continuous and coherent reasoning, making historical feedback more valuable. This discontinuity in NP reasoning limits the effectiveness of historical information for OPT-Agent in such tasks.

## 4 RELATED WORK

### 4.1 LLM EVALUATION

The rapid advancement of Large Language Models (LLMs) has prompted various benchmarks assessing their generalization and reasoning abilities. Early benchmarks like MMLU Hendrycks et al. (2020) and BIG-bench Srivastava et al. (2022) offered broad evaluations but were limited to static, multiple-choice formats, restricting assessment of multi-step reasoning and adaptability. Subsequent benchmarks such as GLUE Wang et al. (2018), SuperGLUE Wang et al. (2019), CommonsenseQA Talmor et al. (2019), HellaSwag Zellers et al. (2019), and TruthfulQA Lin et al. (2022) focused on linguistic and commonsense reasoning but remained single-turn and format-constrained. Math and code benchmarks like MATH Hendrycks et al. (2021), GSM8K Cobbe et al. (2021), HumanEval Chen et al. (2021), and MBPP Austin et al. (2021) target multi-step deduction and synthesis, yet rely on static datasets and one-shot evaluation. Chain-of-Thought prompting Wei et al. (2022) exposes intermediate reasoning but does not transform the static nature of these benchmarks.

Despite these advances, most benchmarks lack dynamic feedback, iterative problem-solving, and long-horizon adaptation—critical for evaluating agent capabilities in large search spaces. NPHardEval Fan et al. (2023) introduces complexity-aware, refreshable tasks across P, NP-complete, and NP-hard classes with automated evaluation, advancing algorithmic generalization assessment. However, it remains confined to static, single-shot paradigms and does not address continual learning or exploration strategies essential for optimization-driven agents.

### 4.2 LLM AGENTS EVALUATION

Recent advances have empowered Large Language Models (LLMs) to act as autonomous agents with multi-step reasoning, tool use, and iterative self-improvement. Frameworks like ReAct Yao et al. (2022a), Toolformer Schick et al. (2023), and ART Parisi et al. (2023) combine decision-making with external tool interaction, while Tree of Thoughts Yao et al. (2023) enhances planning through structured exploration. Reflexion Shinn et al. (2023) and Self-Refine Madaan et al. (2023) incorporate feedback-driven learning to iteratively improve behavior. Benchmarks such as AgentBench Liu et al. (2023) and ToolBench OpenBMB (2023) assess general agent performance and tool usage, whereas domain-specific evaluations like WebShop Yao et al. (2022b), WebArena Zhou et al. (2023), ALFWorld Shridhar et al. (2020), InterCode Yang et al. (2023), and MLE-Bench OpenAI (2024) focus on specialized tasks including web navigation, embodied interaction, interactive coding, and machine learning workflows.

Nonetheless, few benchmarks target LLM agents in large-scale combinatorial optimization. NPHardEval Fan et al. (2023) presents complex algorithmic challenges but lacks iterative feedback and long-horizon optimization. IOLBench Zhang et al. (2024) emphasizes linguistic reasoning without action planning. Existing benchmarks often prioritize task completion or tool execution over sustained optimization and strategic search, underscoring the need for dedicated benchmarks like OPT-BENCH.

## 5 CONCLUSION

We introduce OPT-BENCH, a benchmark comprising 20 ML and 10 NP problems, designed to evaluate LLMs' capabilities in large-scale optimization. Additionally, we present OPT-Agent, which simulates human reasoning by enabling LLMs to iteratively improve solutions using historical feedback. Experiments on 17 state-of-the-art LLMs from 7 families—including reasoning models, general models, and open-source models ranging from 3B to 72B parameters—reveal that leveraging historical context consistently enhances performance across both ML and NP tasks. We find that iteration steps and temperature play a crucial role in convergence and stability. However, the benefit of historical information is limited, and optimization performance varies between reasoning and general models. Reasoning models typically provide better solutions earlier in the process, while open-source models start with lower-quality solutions but improve with more iterations. Open-source models show higher error rates and underperform reasoning models on NP reasoning tasks, indicating room for improvement. Despite these advancements, LLMs still lag behind human experts, demonstrating the ongoing gap between AI and human optimization capabilities. Overall, OPT-BENCH and OPT-Agent provide a comprehensive platform for advancing LLM-driven optimization in real-world challenges.

## REPRODUCIBILITY STATEMENT

We follow the reproducibility guidelines in the ICLR 2026 author guidelines. We will open source all data, and code to reproduce our results as soon as possible.

## ETHICS STATEMENT

The OPT-BENCH dataset was constructed using publicly available sources (e.g., Kaggle) and a rule-based generation program for NP tasks. All privacy-sensitive personal information was removed during the data curation process. To prevent potential misuse, the benchmark will be released under a restrictive license for academic research purposes only, explicitly prohibiting malicious applications.

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

# A  APPENDIX

## A.1  USE OF LARGE LANGUAGE MODELS

Large Language Models are used for grammar check and polishing in this paper.

## A.2  LIMITATIONS

Evaluating large language models on ML optimization tasks is challenged by the growing in-context prompt size, as accumulated code and historical data expand the context window, raising evaluation costs and nearing token limits. To address this, we seek methods to reduce the context window size, enabling more optimization steps while maintaining efficiency and scalability. Moreover, since OPT-BENCH-ML encompasses diverse Kaggle competitions across various domains, averaging performance metrics may introduce scale inconsistencies.

## A.3  DRAFT SETTING

Unlike the refine setting, the draft setting requires models to generate solutions from scratch, providing a more direct test of their optimization capabilities. Results for three strong models are shown in Table 6. Notably, the open-source `Qwen2.5-72B-Instruct` consistently shows higher buggy rates than proprietary models, reflecting greater difficulty in producing valid solutions during draft optimization. However, except for `grok-3` at 5 steps, all models achieve higher improvement rates ($IR(d,r)$) than in the refine setting, indicating that draft optimization can outperform traditional refinement when valid solutions are found. The *Win Count* metric further shows that using historical information during draft optimization improves performance across all step counts. Increasing the number of steps yields significant gains in both improvement rates and win counts, highlighting the value of iterative refinement. These results emphasize the importance of managing the trade-off between exploration and solution validity in draft optimization and suggest that reducing buggy rates is key to advancing both proprietary and open-source LLMs in this setting.

| Model | 5 steps | | | | 10 steps | | | | 20 steps | | | |
|---|---|---|---|---|---|---|---|---|---|---|---|---|
| | Buggy Rate | | Win Count | IR(d,r) | Buggy Rate | | Win Count | IR(d,r) | Buggy Rate | | Win Count | IR(d,r) |
| | w | w.o | | | w | w.o | | | w | w.o | | |
| gpt-4o-2024-08-06 | 0.20 | 0.20 | 8/6 | 1.24 | **0.15** | **0.15** | **10/5** | 1.38 | **0.15** | **0.15** | **11/4** | 1.41 |
| grok-3 | **0.15** | **0.15** | **10/5** | 0.80 | **0.15** | **0.15** | **10/5** | 1.38 | 0.15 | 0.15 | **11/4** | **1.42** |
| Qwen2.5-72B-Instruct | 0.40 | 0.35 | 5/5 | **1.54** | 0.30 | 0.30 | 6/6 | **1.94** | 0.20 | 0.20 | 8/7 | 1.12 |

Table 6: **Evaluation Results of LLMs under Draft Settings.** Metrics include *Buggy Rate*, denoting the proportion of invalid solutions; *Win Count*, comparing OPT-Agent-draft optimization against the baseline without historical information; and *IR(d,r)*, the improvement rate comparing OPT-Agent-draft optimization to OPT-Agent-refine.

## A.4  OPT-BENCH DATASET

The detailed information regarding the 20 Machine Learning (ML) tasks and 10 NP problems used in our OPT-BENCH is comprehensively summarized in Table 7 and Table 8, respectively. These tables provide concise descriptions of each task or problem, along with their corresponding evaluation metrics, which serve as the foundation for assessing the performance of OPT-Agent across diverse optimization scenarios.

# B  OPT-AGENT PROMPT

In this section, as illustrated in Figure 4 and Figure 7, we provide a comprehensive overview of the prompt templates used in both OPT-Agent-ML and OPT-Agent-NP. These prompts guide the model through three distinct types of actions—draft, improve, and debug—by delivering task-specific context and structured response formats. Specifically, the OPT-Agent-ML prompts focus on instructing the model for machine learning tasks, while the OPT-Agent-NP prompts are carefully designed to include structured task descriptions, input-output examples, and response formatting guidelines, enabling the model to systematically address and refine solutions to NP problems.

| Kaggle Competition | Description | Metric |
|---|---|---|
| bike-sharing-demand | Forecast use of a city bikeshare system | Root Mean Squared Logarithmic Error (RMSLE) ↓ |
| competitive-data-science-predict-future-sales | Predict total sales for every product and store | Root Mean Squared Error (RMSE) ↓ |
| house-prices-advanced-regression-techniques | Predict house sales prices | Root-Mean-Squared-Error (RMSE) ↓ |
| london-house-price-prediction-advanced-techniques | Predict London house prices | Mean Absolute Error (MAE) ↓ |
| playground-series-s3e14 | Predicting wild blueberry yields | Mean Absolute Error (MAE) ↓ |
| playground-series-s3e16 | Predict the age of crabs | Mean Absolute Error (MAE) ↓ |
| playground-series-s3e19 | Forecast Mini-Course Sales | Symmetric Mean Absolute Percentage Error (SMAPE) ↓ |
| playground-series-s3e22 | Predict Health Outcomes of Horses | micro-averaged F1-Score ↑ |
| playground-series-s3e24 | Binary Prediction of Smoker Status using Bio-Signals | area under the ROC curve ↑ |
| playground-series-s3e25 | Regression with a Mohs Hardness Dataset | Median Absolute Error (MedAE) ↓ |
| playground-series-s3e3 | Binary Classification with a Tabular Employee Attrition Dataset | area under the ROC curve ↑ |
| playground-series-s3e5 | Ordinal Regression with a Tabular Wine Quality Dataset | quadratic weighted kappa ↑ |
| playground-series-s4e2 | Multi-Class Prediction of Obesity Risk | Accuracy ↑ |
| sentiment-analysis-on-movie-reviews | Classify the sentiment of sentences from the Rotten Tomatoes dataset | classification accuracy ↑ |
| spaceship-titanic | Predict which passengers are transported to an alternate dimension | classification accuracy ↑ |
| tabular-playground-series-aug-2022 | Predict product failures | area under the ROC curve ↑ |
| tabular-playground-series-feb-2021 | Predict the amount of an insurance claim | Root-Mean-Squared-Error (RMSE) ↓ |
| tabular-playground-series-jul-2021 | Predict air pollution in a city | Root Mean Squared Logarithmic Error (RMSLE) ↓ |
| tabular-playground-series-sep-2022 | Predict book sales | Symmetric Mean Absolute Percentage Error (SMAPE) ↓ |
| telstra-recruiting-network | Predict the severity of service disruptions on their network | multi-class logarithmic loss ↓ |

Table 7: **Kaggle Machine Learning Competitions with Description and Metric.**

| NP Problem | Description | Metric |
|---|---|---|
| Graph Coloring Problem (GCP) | Use the minimum number of colors necessary to achieve a valid coloring | Color Number ↓ |
| Hamiltonian Cycle | Find the largest possible valid Hamiltonian circuit | Path Length ↑ |
| Knapsack | Choose a subset of items to pack into a limited-capacity bag | Total Item Weight ↑ |
| Maximum Clique Problem | Find the largest clique (a subset of vertices all connected to each other) in a given graph | Clique Size ↑ |
| Maximum Set | Find the largest subset of a set under constraints | Set Size ↑ |
| Meeting Schedule | Schedule meetings for as many as participants under various constraints | Total Attendees ↑ |
| Minimum Cut | Find the minimum cut in a graph | Cut Weight ↓ |
| Set Cover | Select a minimal number of sets from a collection such that their union covers all elements of a universal set | Subset Number ↓ |
| Subset Sum | Find a subset of a set of numbers that adds up to a specific target value | Indice Number ↑ |
| Traveling Salesman Problem (TSP) | Find the shortest possible route that visits each city once and returns to the starting point | Route Length ↓ |

Table 8: **NP Problems with Description and Metric.**

## B.1 OPT-AGENT RESULTS ANALYSIS

## B.2 ML TASK

As shown in Figure 5, we present the optimization trajectory of OPT-Agent tackling bike sharing demand ML task, illustrating progressive improvements in evaluation metrics through various strategies. Beginning with hyperparameter tuning and sampling enhancements, the model undergoes iterative refinements including the introduction of early stopping, regularization techniques, and learning rate adjustments. The diagram also highlights encountered issues, such as the early stopping rounds exception, along with the corresponding fixes, demonstrating a systematic approach to model optimization and performance enhancement.

## B.3 NP PROBLEM

As shown in Figure 6, we illustrate the solution refinement process of OPT-Agent applied to the Hamiltonian Cycle NP problem. The flowchart depicts iterative attempts to build a valid Hamiltonian circuit by resolving challenges such as disconnected nodes and repeated visits. Each step includes metric evaluations, detailed state information, and proposed paths, demonstrating how OPT-Agent systematically enhances the solution toward a valid and optimized Hamiltonian cycle.

---

**OPT-Agent-ML**

**Introduction (draft):** You are a Kaggle grandmaster attending a competition <task type>. In order to win this competition, you need to come up with an exceptional and creative plan. To address this problem, I will provide you with the specific task description, the evaluation metrics to be used, training set and submission format in sequence.

**Introduction (improve):** You are a Kaggle grandmaster attending a competition <task type>. You have been provided with previously developed solution, and your task is to improve it in order to increase the performance in test dataset. Review previous solution and improve based on it. You can only modify the model, optimizer, or hyperparameters, and adjust feature engineering for compatibility.

**Introduction (draft):** You are a Kaggle grandmaster attending a competition <task type>. The previous solution contains a bug. According to the buggy information, revise it to fix the issue.

**Task description:** <task description>

**Evaluation metric:** <metric>

**Training set format:** <dataset description>

**Submission format:** <submission format>

**History Information:** <history information>

**Previous Solution:** <previous solution>

**Previous (buggy) Implementation:** <previous (buggy) implementation >

**Previous (buggy) Output:** <previous (buggy) output >

**Instructions:** <Response Format>, <Implementation Guideline>, <Solution Draft Sketch Guideline>, <Solution Improvement Sketch Guideline>, <Solution Debug Sketch Guideline>

---

**OPT-Agent-NP**

**Introduction (draft):** You are a great expert solving <task description> question. You should propose a solution to this question.

**Introduction (improve):** You are a great expert solving <task description> question. You should optimize the solution based on the history information.

**Introduction (debug):** You are a great expert solving <task description> question. You should debug the solution based on the previous buggy information.

**Task description:** <task description>

**Submission Format:** <submission format>

**Question:** The <task type> question is: <question>

**History Information:** <history information>

**Previous Buggy Information:** <Previous buggy information>

**Example Input and Output:** <Example Input and Output>

**Instructions:** <Instructions>

**Response Format:** <Response Format>

---

Figure 4: **Prompt Template of OPT-Agent.** Orange denotes draft action. Green denotes improve action. Purple denotes debug action. Blue denotes shared prompts.

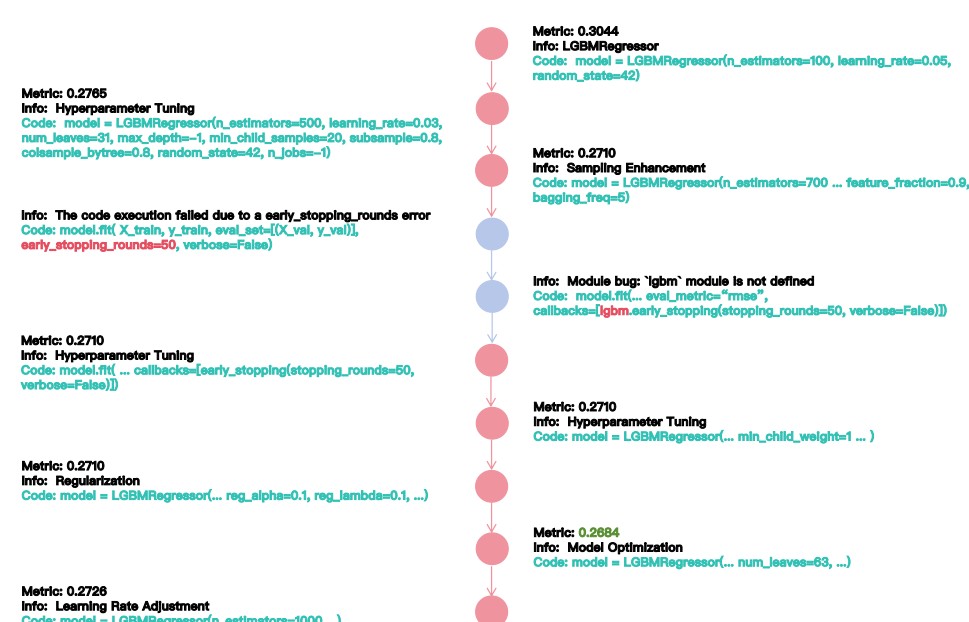

Figure 5: **Detailed OPT-Agent-ML Trace on the Bike Sharing Demand Task**, utilizing `gemini-2.0-flash` as LLM base model. The red, and blue nodes represent the improve, and debug action, respectively.

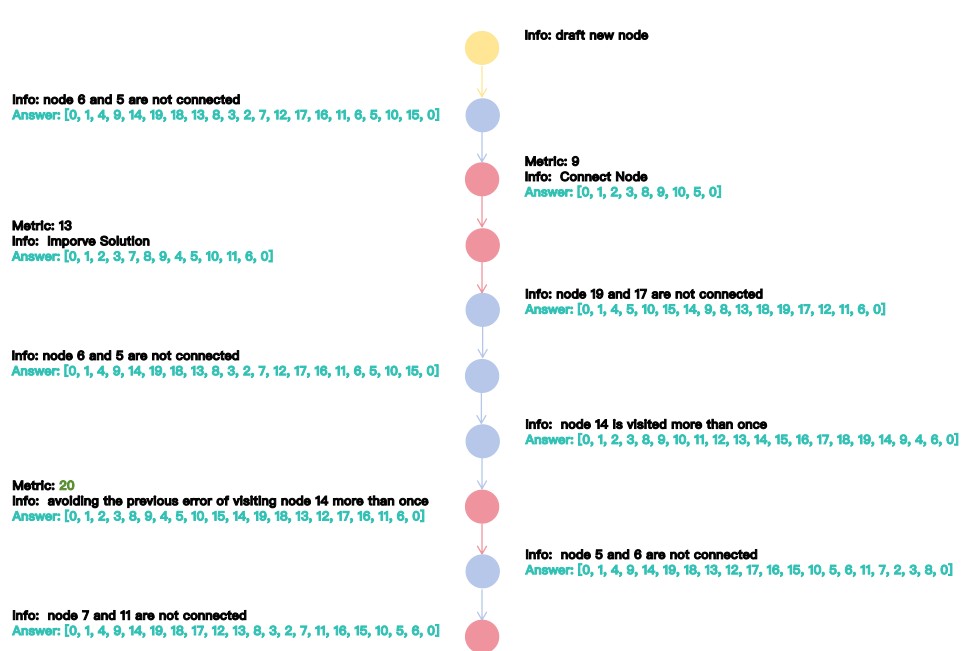

Figure 6: **Detailed OPT-Agent-NP Trace on the Hamiltonian Cycle Task**, utilizing `gemini-2.0-flash` as the LLM base model. The yellow, red, and blue nodes represent the draft, improve, and debug action, respectively.

**OPT-Agent-ML**

**Response format:** Your response should be a brief outline/sketch of your proposed solution for model, optimizer, and hyperparameters selection in natural language (3-5 sentences), followed by a single markdown code block (wrapped in "') which implements this solution and prints out the evaluation metric. There should be no additional headings or text in your response. Just natural language text followed by a newline and then the markdown code block. Note that the code block should be a complete Python program.

**Impl guideline:** Be aware of the running time of the code, it should complete within `time`. All data is already available in the `./input` directory. You can also use the `./working` directory to store any temporary files that your code needs to create. The evaluation should be based on `k-fold-validation` but only if that's an appropriate evaluation for the task at hand.

**Solution draft sketch guideline:** The initial solution design should be simple, efficient, and avoid overfitting, with minimal iterations. Take the Memory section into consideration when proposing the design, and do not propose the same modeling solution while keeping the evaluation the same. The solution sketch should be 3-5 sentences and propose a reasonable evaluation metric for this task. Do not suggest performing Exploratory Data Analysis (EDA). The data is already prepared and available in the `./input directory`, so there is no need to unzip any files. Note that the training dataset should be shuffled before splitting into training and validation sets, and the random seed (state) should be fixed.

**Solution improvement sketch guideline:** The solution sketch should be a brief natural language description of how the previous solution can be improved. You should be very specific and propose only a single actionable improvement. Do not suggest performing Exploratory Data Analysis (EDA). Ensure that function parameters match the official documentation by checking for accuracy, compatibility, and any deprecated or renamed parameters, referring to the latest examples if needed. Note that only the model, optimizer, hyperparameters, and feature engineering should be modified. This improvement should be atomic so that its effect can be experimentally evaluated. Additionally, take the Memory section into consideration when proposing the improvement. The solution sketch should be 3-5 sentences.

**Solution debug sketch guideline:** You should write a brief natural language description (3-5 sentences) of how the issue in the previous implementation can be fixed. Do not suggest performing Exploratory Data Analysis (EDA). Ensure that function parameters match the official documentation by checking for accuracy, compatibility, and any deprecated or renamed parameters, referring to the latest examples if needed. If the previous buggy solution was due to time limitations, focus on reducing the code's time consumption rather than fixing the bug—for example, by simplifying the model's hyperparameters, reducing the number of iterations, or switching from K-Fold cross-validation to a single train-test split. Additionally, take the Memory section into consideration when proposing the improvement.

**OPT-Agent-NP**

**Example Input and Output:** Here is the example input and output: Input: <example input> Output: <example output>.

**Instructions:** You should only output the answer of this task following Submission Format. Do not output code or any explanation. The output must not include anything other than the final answer. Ensure your response stays within the maximum token limit. Avoid repeating or padding the output.

**Response Format:** The last part of your response should be of the following format: Answer: <YOUR ANSWER> (without angle brackets) where YOUR ANSWER is your answer. For your answer, do not output additional contents that violate the specified format.

Figure 7: **Fixed prompts in OPT-Agent**. This encompasses the response format, implementation guidelines, solution draft sketch guidelines, solution improvement sketch guidelines, and solution debug sketch guidelines for ML tasks, as well as example inputs and outputs, instructions, and response format for NP problems.

