# OpenReview forum: "OPT-BENCH: Evaluating LLM Agent on Large-Scale Search Spaces Optimization Problems"
_ICLR.cc/2026/Conference — ICLR 2026 Conference Withdrawn Submission_

### Official Review · Reviewer_8LdH · 2025-10-27

**Soundness:** 3
**Presentation:** 3
**Contribution:** 3
**Rating:** 6
**Confidence:** 4

**Summary:**

This paper introduces OPT-BENCH, a comprehensive benchmark designed to evaluate Large Language Model (LLM) agents on large-scale search space optimization problems. OPT-BENCH includes 20 real-world machine learning (ML) tasks sourced from Kaggle and 10 classical NP problems, providing a diverse and challenging environment for assessing LLMs on iterative reasoning and solution refinement. The authors also present OPT-Agent, an end-to-end optimization framework that emulates human reasoning by generating, validating, and iteratively improving solutions using historical feedback. Extensive experiments with 17 state-of-the-art LLMs from 7 model families demonstrate that incorporating historical context enhances optimization performance, though a gap remains compared to human experts. All datasets, code, and evaluation tools will be open-sourced to foster further research.

**Strengths:**

- The benchmark covers both ML and NP-hard problems, with clear task definitions, evaluation metrics, and human expert baselines.
- OPT-Agent’s workflow (draft, improve, debug) closely mirrors human iterative problem-solving.
- The experiments are extensive, covering a wide range of LLMs (proprietary and open-source, 3B–72B parameters), and include ablation studies on temperature and optimization steps.
- Results are reported with multiple metrics (Win Count, Buggy Rate, Average Ratio, Improvement Rate), and the analysis is detailed and nuanced.
- The limitations and reproducibility are transparently discussed.

**Weaknesses:**

- While OPT-Agent is well-implemented, its core workflow (draft, improve, debug) is conceptually similar to existing agent frameworks. The main novelty lies in the benchmark and evaluation protocol.
- The paper notes that historical feedback is less effective for NP problems, as LLMs often fail to incrementally refine solutions and instead generate new ones. More analysis or proposed solutions for this limitation would strengthen the work.
- As acknowledged, averaging performance across diverse ML tasks may introduce scale inconsistencies. More discussion on normalization or task weighting could be helpful.
- The growing context window in iterative optimization increases evaluation costs and may hit token limits. While this is discussed, concrete solutions or future directions would be valuable.


### Typos
Table 3 caption should be OPT-BENCH-NP

**Questions:**

- Is there a better way to ablate the impact of historical feedback? There are many discussion about the impact historical feedback but the detail of how historical feedback is incorporate is not revealed. E.g. compare with no historical feedback version of OPT-agent.

---

### Official Review · Reviewer_iJ6Q · 2025-10-30

**Soundness:** 2
**Presentation:** 3
**Contribution:** 2
**Rating:** 2
**Confidence:** 4

**Summary:**

This paper aims to address a key gap in LLM evaluation: the lack of benchmarks for assessing their ability to iteratively optimize complex solutions by learning from historical feedback, a core component of human problem-solving. First, they introduce OPT-BENCH, a2e4 new benchmark comprising 30 tasks (20 real-world ML problems from Kaggle and 10 classical NP-hard problems) designed specifically to test this iterative refinement capability. Second, they propose OPT-Agent, an end-to-end framework that enables LLM agents to generate, validate, and iteratively improve solutions by leveraging feedback from previous attempts.

**Strengths:**

1. The paper addresses an important problem. Most current LLM benchmarks focus on single-turn, static QA or reasoning. Evaluating LLMs on complex, long-horizon optimization tasks that require iterative refinement and learning from historical feedback is a critical next step for LLM agent research.
2. OPT-BENCH's feature lies in its combination of two distinct but challenging domains: real-world ML problems (requiring code generation, hyperparameter tuning, and data understanding) and classical NP problems (requiring pure logical and combinatorial reasoning). This composition provides a diverse perspective for evaluating LLMs.
3. The paper provides a "human expert" baseline for both ML tasks (Kaggle gold medal solutions) and NP problems (approximate optimal solutions from heuristic algorithms). It makes the "Average Ratio (AR)" metric meaningful and quantifies the true gap between LLMs and high-level human performance.
4. It proves that historical feedback does improve performance (albeit limitedly). It quantifies the significant gap between LLMs and human experts. It reveals the "brittleness" of LLMs in handling NP problems (i.e., difficulty in incremental repair, preferring to start over completely), which is a very specific and insightful finding.

**Weaknesses:**

1. One of the paper's core contributions, OPT-BENCH, is essentially a collection and reformatting of existing problems (Kaggle, classical NP problems), rather than the creation of new, specifically designed evaluation tasks.
2. Compared to benchmarks like MLE-Bench (75 tasks), the number of ML tasks in OPT-BENCH (20) seems small. More importantly, the paper does not adequately justify the representativeness of these 20 tasks. Are they primarily biased towards tabular data? Do they cover other ML domains requiring complex optimization? Without this comprehensiveness, the benchmark's evaluation conclusions may be skewed.
3. The OPT-Agent framework itself appears to be a standard "reflect-debug-improve" loop, lacking significant structural innovation.
4. The field already has extensive research on LLM agent reflection mechanisms and thinking strategies. The paper fails to conduct a horizontal comparison of OPT-Agent with these SOTA agent frameworks. This lack of comparison makes the attribution of results unclear. For instance, is the poor performance on NP problems due to the LLM's (e.g., GPT-4) limited reasoning ability, or is it because the simple reflection mechanism of OPT-Agent is inefficient? Would the LLM's performance be drastically different with a stronger agent framework?
5. The paper's evaluation framework relies on several metrics (Win Count, Buggy Rate, IR, AR), but their justification, independence, and comprehensiveness are unclear.
 (1) Win Count: This metric primarily seems to function as an ablation study demonstrating that historical information is beneficial. Could the authors elaborate on what additional value this metric provides for comparing the relative optimization capabilities of different models (e.g., GPT-4 vs. Claude-3.5)?
 (2) Metric Suite: The paper proposes four metrics. Could the authors clarify which distinct capability each metric is intended to measure? For example, Buggy Rate appears to measure reliability, while AR measures final solution quality. What unique insights do IR and Win Count provide that are not captured by the others?
 (3) Missing Practical Metrics: The evaluation of an "optimization" task seems incomplete without considering the cost of finding the solution. Why did the authors choose to omit crucial practical metrics like computational overhead (e.g., response time per iteration, total API token consumption/cost)? Without this data, isn't it difficult to assess the practical trade-offs (e.g., solution quality vs. cost) between different models, which is a key aspect of real-world optimization?

**Questions:**

1. Why did you choose to include only 20 ML tasks, rather than leveraging a larger, more established set of ML tasks like MLE-Bench (75 tasks)? How can you justify that these 20 tasks are comprehensive enough to evaluate LLM optimization capabilities across various ML domains, and not just biased towards tabular data?
2. The contribution of the benchmark seems to be mainly "collection." Compared to other benchmarks that also evaluate LLM problem-solving, is the core advantage of OPT-BENCH the focus on "iteration," or the specific inclusion of "NP problems"?
3.  Can you clarify the fundamental design difference between OPT-Agent and existing reflection frameworks (like Reflexion)? It appears to follow a standard reflect-debug loop.
4.  Why did the paper not compare OPT-Agent with any other SOTA agent frameworks? Without such a comparison, how can you be sure that the performance bottlenecks observed (especially on NP problems) stem from the LLM's intrinsic limitations, and not from potential inefficiencies in the OPT-Agent framework itself?
5.  The "Win Count" metric seems to just validate your hypothesis (that history is beneficial). What additional value does it provide for comparing the optimization capabilities of different LLMs (e.g., GPT-4 vs. Claude-3.5)?
6.  Could you provide data on the models' computational cost, such as average response time per iteration, total computation time, or API costs? Without this data, the assessment of "optimization" performance seems incomplete, as it ignores the practical trade-offs between solution quality and cost.

---

### Official Review · Reviewer_oFSj · 2025-10-31

**Soundness:** 2
**Presentation:** 3
**Contribution:** 2
**Rating:** 2
**Confidence:** 3

**Summary:**

This paper introduces OPT-BENCH, a benchmark designed to evaluate the ability of LLMs to solve large-scale search space optimization problems through iterative refinement. The benchmark is composed of real-world ML tasks sourced from Kaggle and a set of classical NP-hard problems. In conjunction with the benchmark, the authors propose OPT-Agent, an end-to-end framework that allows an LLM agent to generate, validate, and iteratively improve solutions by incorporating historical feedback, thereby emulating human reasoning. The study evaluates a range of state-of-the-art LLMs using this framework to analyze the impact of historical context on  optimization performance and compares the outcomes to human expert baselines.

**Strengths:**

1. The paper attempts to address an underexplored area: the ability of LLMs to iteratively refine solutions based on historical feedback , moving beyond static, single-shot evaluations. This focus on learning from both successes and failures over time is a relevant research direction, as it aims to evaluate a more complex aspect of reasoning that many current benchmarks overlook.

2. The authors have curated a new benchmark, OPT-BENCH, which organizes 20 ML tasks and 10 NP problems into a structured format for evaluating this iterative capability .

3. The paper provides an empirical study across a broad range of current LLMs (17 models from 7 families), including proprietary, open-source, and reasoning-specific models. This testing offers initial data on how different models perform within the proposed optimization framework and analyzes factors like the impact of historical information and iteration count .

**Weaknesses:**

1. The benchmark's scale is too small for statistical significance. With only 20 ML tasks and 10 NP problems (each with just 5 instances), the benchmark lacks the scale to draw reliable conclusions. This problem is exacerbated by the high heterogeneity of the ML tasks, which use disparate metrics (e.g., RMSE, MAE, ROC AUC). As the authors rightly admit in the appendix, averaging these metrics "may introduce scale inconsistencies".


2. A Flawed Evaluation Paradigm for NP Problems: The paper tasks LLMs with searching directly within the vast solution space (e.g., generating a node path), a flawed paradigm given the combinatorial complexity. A more reasonable approach, consistent with the ML tasks, would be to evaluate the LLM's ability to design or refine an algorithm (e.g., write heuristic code). The paper's own findings—that LLMs "generate a completely new solution" instead of applying incremental feedback—confirm that direct search is an ineffective task for them


3. The "human expert" baseline is inconsistent and weak. The ML baseline is a Kaggle "gold medal solution," which is a strong, appropriate standard. However, the NP baseline is just "a heuristic algorithm," which is a very weak proxy for expert performance (e.g., compared to a professional solver) and makes the performance gap analysis for NP tasks unreliable.

**Questions:**

1 Given the findings that LLMs struggle to apply incremental feedback to direct solutions for NP problems, did you consider an alternative paradigm? Specifically, why not evaluate the LLM's ability to generate or refine an algorithm (e.g., heuristic code), which seems more analogous to the ML tasks?
2 Could you justify the choice of "a heuristic algorithm" as the "human expert" baseline for NP tasks, especially when a much stronger baseline (e.g., an optimal solution from a professional solver) is often available? How might this weak baseline affect your conclusions about the performance gap?
3  Regarding the 20 ML tasks, the metrics are highly heterogeneous. How do you account for the "scale inconsistencies" when drawing general conclusions, and how does the small number of tasks support the statistical significance of the results?

---

### Official Review · Reviewer_GRv5 · 2025-11-01

**Soundness:** 2
**Presentation:** 3
**Contribution:** 2
**Rating:** 2
**Confidence:** 4

**Summary:**

The paper introduces OPT-BENCH benchmark to evaluate the performance of LLM agents on real-world ML tasks and NP problems. It also proposes OPT-Agent, which uses the historical context to debug and improve the solutions. The experiments cover 17 LLMs from different model families, which highlight the performance gap comparing to human experts.

**Strengths:**

The paper introduces OPT-Agent, an evaluation pipeline that mimics how modern LLM agents actually operate. Comparing to the single-pass inference setups used by many other benchmarks, OPT-Agent enables multi-round refinement by feeding historical feedback and prior attempts back into the context. This design captures agentic problem solving and generates a more informative evaluation.

**Weaknesses:**

1. Limited contributions of the benchmark. For the ML tasks, OPT-BENCH-ML looks similar to MLE-bench, where the metrics focus on the comparison to human expert and performance on the test set. It does not provide the intrinsic explanations for the LLM performance on ML tasks. For example, if the OPT-BENCH-ML wants to provide more explainability, it may focus on the diversity of proposed ML solutions, success rate of correctly implementing a ML algorithm, etc. These low-level evaluations are more helpful to understand and analyze the performance gap.
2. Data leakage risk from raw Kaggle datasets. Evaluating directly on public Kaggle datasets introduces leakage risks, especially for recent frontier models likely exposed to leaderboard code.
3. The LLMs used in the experiments do not include the SOTA models on coding tasks, such as claude-4-sonnet, GPT5 and gemini-2.5pro.

**Questions:**

It would be helpful to address the weaknesses above, through providing new metrics, evaluating the data leakage and including SOTA models on coding tasks.

---

### Note · Authors · 2026-01-08

I have read and agree with the venue's withdrawal policy on behalf of myself and my co-authors.